# Mechanism of Morphology Development in HDGEBA/PAMS Hybrid Thermosets: Monte Carlo Simulation and LSCM Study

**DOI:** 10.3390/polym14245375

**Published:** 2022-12-08

**Authors:** María G. González, Javier Pozuelo, Juan C. Cabanelas, María B. Serrano, Juan Baselga

**Affiliations:** Department of Materials Science and Engineering and Chemical Engineering, IAAB, University Carlos III of Madrid, 28911 Leganés, Spain

**Keywords:** hydrogenated glycidyl ether of bisphenol A, poly(3-aminopropylmethyl-siloxane), LSCM, Monte Carlo simulation

## Abstract

Reactive combinations of aliphatic epoxy resins and functional polysiloxanes form a class of hybrid thermosetting materials with properties that may come from both the organic and the inorganic phases. The two typically immiscible phases form a suspension whose morphology, composition, and thermal properties vary with curing time. The aim of this research was to elucidate the mechanism by which morphology changed with time and to simulate it through Metropolis-Monte Carlo. The selected system was hydrogenated epoxy (HDGEBA) and a synthetic polyaminosiloxane (PAMS). It was studied by DSC, FTnIR, gel point, viscometry, and in-situ laser scanning confocal microscopy. A mechanism for morphology generation was proposed and simulated, exploring a wide range of values of the “a priori” relevant variables. The essential features were captured by simulations with a reasonable agreement with experimental data. However, the complete process was more complex than the geometrical approach of the simulation. The main deviations that were found and qualitatively explained are: (i) the induction period on the rate of coalescence, and (ii) PAMS-rich domain average size increases faster than predictions.

## 1. Introduction

Epoxy resins are widely used as adhesives, coatings, and matrices in polymer composites due to their low viscosity, good insulating properties even at high temperatures, and good chemical and thermal resistance [1,2]. Nevertheless, this kind of material demonstrates two important limitations: they are brittle (due to their high crosslinking degree) [3] and have high water uptake [4]. To improve toughness, epoxy resins are commonly modified with reinforcements of a different nature (elastomers [5], thermoplastics [6], inorganic particles [7]), geometry (particles, fibers, platelets) [8,9,10,11], and size (micro and nano) [12,13,14,15], which provide additional energy absorption mechanisms [16,17,18,19,20].

The preparation of hybrid materials incorporating siloxanes, which are highly hydrophobic [21,22,23,24], into the epoxy formulation is a promising route to the development of advanced epoxies. Polysiloxanes can impart high thermal, chemical, and photochemical stability and can reduce the water uptake [25], as well as the surface tension, improving wetting against low energy substrates.

Nevertheless, the low miscibility of siloxanes with epoxy resins promotes its segregation in the early stage of the curing process [26,27], preventing a good dispersion of the organo-inorganic phase. Many strategies have been explored during the last few years, including the synthesis of graft copolymers with highly soluble polymers [28,29] and the use of end-functional silicones with different groups (epoxy, amine, acrylate, silanol, isocyanate, etc.) [30,31], to improve miscibility and optimize the phase segregation process. The use of reactive polysiloxanes, where the siloxane chemically bonds to the matrix, partially solves the problem. A reactive polymer mixture that initially forms a single homogeneous phase is entropically driven to separate into chemically distinct phases due to the increase of molecular weight associated with step-growth polymerizations. However, when employing immiscible components like reactive polysiloxanes, in order to obtain the desired level of properties and performance, compatibilization methods are needed via, for example, incorporation of a reactive compatibilizer or via self-compatibilization by chemical reaction between components. Though the initial mixture is heterogeneous, the graft copolymer formed during curing acts as a compatibilizer. Depending on functionality degree, the siloxane may remain dissolved in the matrix or may undergo phase separation [32,33,34]. In most cases, at least compositional gradients are observed in the cured material [35,36,37], though some homogeneous materials have been developed [38,39,40].

In this study, a synthetic multifunctional siloxane, poly(3-aminopropylmethylsiloxane) (PAMS, Table 1) was used as a hardener for an aliphatic epoxy resin. This hardener has been previously studied in DGEBA-based systems [25,27]; it was initially immiscible and a complex reactive compatibilization process seems to give rise to quasi-homogenous fully-cured materials, but the question about the morphology evolution during curing remains open. Mapping of silicon content at the surface suggested the existence of macrodomains [25,27,41] with a size of tens of microns in a certain range of curing temperatures and conversions. Laser scanning confocal microscopy (LSCM) studies of DGEBA/PAMS cured up to limiting conversion confirmed the presence of siloxane-rich domains at curing temperatures between 21 °C and 120 °C, and the presence of compositional gradients between the matrix and the domains, which are temperature dependent. Images obtained during the curing of this reactive system suggest three main mechanisms for morphology development: coalescence, diffusion, and chemical reaction [27,35].

This work aims to give a deeper insight into the processes which generate the final morphology of this kind of reactive systems. But PAMS is highly reactive against DGEBA, and this fact prevents careful experiments from being done to elucidate the mechanisms of morphology evolution during curing. In this work, a less reactive aliphatic epoxy resin has been selected and using LSCM and image techniques, it has been possible to propose a mesoscopic model of network formation.

## 2. Materials and Methods

### 2.1. Samples Preparation

Table 1 shows the main characteristics of the monomers and polymers used in this work. The hydrogenated derivative of diglycidyl ether of bisphenol-A (HDGEBA) was supplied by CVC specialty Chemicals (Maple Shade, NJ, USA), with an equivalent mass of 210 g·mol^−1^, determined by acid titration. The amount of secondary hydroxyl groups was 0.4864 mol·kg^−1^. The curing agent was poly(3-aminopropylmethylsiloxane) (PAMS), obtained according to a previously published method [35]. PAMS was characterized by vapour pressure osmometry (VPO, Knauer K7000); *M_n_* was 1094 ± 64g·mol^−1^ and amine hydrogen equivalent was 58.5 g·mol^−1^, which yields a number average amine hydrogen functionality of 18.7. By MALDI-TOF, it was possible to determine the mass average amine functionality, molar mass, and the presence of a notable amount of cycles (35% *w*/*w*). The PAMS oligomer was stored at 4 °C before use. 

For Laser Scanning Confocal Microscopy (LSCM) studies, PAMS was labeled with an amine reactive fluorescent chromophore, Rhodamine-B sulphonyl chloride (RhB, Lissamine^®^ Molecular Probes). The dye was dissolved in the oligomer and the solution was heated at 60 °C for 4 h in an N_2_ atmosphere. Complete sulphonamide formation was verified by thin layer chromatography. The label concentration in the modified oligomer was 10^−4^ mol·kg^−1^. 

Curing was performed by mixing appropriate amounts of the reactants in a previously degassed 20 mm diameter glass vial, stirring at room temperature with a magnetic stirrer (10 × 2 mm Teflon coated cylindrical magnet) at 100–300 rpm for 2 min (maximal shear strain about 10–30 rad·s^−1^) and curing at the desired temperature on the appropriate preheated device. The equivalent ratio of epoxy/amine was kept constant and equal to one in all cases. The curing temperatures were selected between room temperature and 120 °C.

### 2.2. Measurements

Thermal transitions were determined by differential scanning calorimetry (DSC). A Mettler-Toledo 822 calorimeter equipped with a liquid nitrogen reservoir was used. Scans were performed at 10 °C/min. 

Complex viscosity was measured during curing at temperatures from 40 °C to 70 °C with a Bohling CVR rheometer coupled to a Haake thermostatic oil bath. A 2° and 20 mm cone-plate geometry was used. Oscillatory measurements were performed at 1Hz and low amplitude, ensuring linear behaviour. 

Solubility tests during curing at different temperatures were performed to determine gel conversion. The samples were cured in an oven for different times and then quenched in an excess of chloroform, a good solvent of both the epoxy and polysiloxane components. The quenched samples were stirred for 4 h, and the presence of an insoluble fraction was checked and weighted; microgel conversion was determined by extrapolation to zero weight. 

Blends morphology was studied by laser scanning confocal microscopy (LSCM, Zeiss LSCM-5 Pascal). Excitation was carried out with a 25 mW Ar laser at 488 nm. Two different objectives were used: Achroplan 20×/0.45 (scanned area 460 μm × 460 μm, axial waist radius 5.27 µm, lateral waist radius 0.47 µm) and Ultrafluar 40×/0.6 (immersion objective, scanned area 230 μm × 230 μm, axial waist radius 2.90 µm, lateral waist radius 0.36 µm). In all cases, images were taken at 100 µm from the surface. The samples for LSCM observation were prepared between a glass slide and a glass cover separated by a 1 mm Teflon spacer and cured in an oven at the desired temperature. Images were taken after quenching at room temperature. For the “in situ” study of morphology evolution during curing, the microscope was equipped with a thermostated stage (Linkam Sci. PEI 120). Under the conditions used for the “in situ” study, photodegradation was minimized.

For solubility tests, viscosity measurements, and LSCM observations, it was necessary to determine the chemical conversion at the temperatures and reaction times at which each measurement took place. The conversion was measured by Fourier Transform Infrared Spectroscopy in the near range (FTnIR, Perkin-Elmer GX2000, 4 cm^−1^ resolution). After mixing stoichiometric amounts of both components, the sample was placed on a preheated sample cell formed by two glass slides and a Teflon spacer (0.5 mm thickness), inserted in the spectrometer, and scanned at constant time intervals. The epoxy conversion (α) versus time master curves were determined following the extinction of the epoxy band at 4526 cm^−1^ at each curing temperature. See ref [4] for spectra details.

## 3. Results

### 3.1. Gelation, Viscosity Measurements, and Glass Transition Temperatures

Homogeneous monomeric epoxy systems usually follow the Stockmayer gelation theory. For epoxy–amine systems in which one or both components present a distribution of molecular masses, gel conversion can be calculated according to [42] $\alpha_{G}^{2}={\left[\left(f_{wE}-1\right)\left(f_{wA}-1\right)\right]}^{-1}$, where $f_{wE}$ and $f_{wA}$ are the mass average functionality of the epoxy and the amine components, respectively. This equation predicts, in our case, a gel conversion of 0.23 ± 0.02. Nevertheless, the appearance of the first insoluble fraction as measured by solubility tests takes place at α=0.15±0.01, irrespective of the curing temperature (see Appendix A). This small difference may be attributed to experimental errors in the characterization of PAMS functionality, although it may be possible that the cyclic fraction of PAMS (see experimental part) may bring forward gelation. This last hypothesis cannot be confirmed yet due to the lack of an appropriate theory for the gelation of cyclic monomers. 

Complementary viscosity measurements (Appendix A) were obtained to verify the percolation threshold, and it was found that the conversion at which complex viscosity diverges is temperature dependent and almost doubles the experimental gel conversion (Table 2); as the temperature increases, percolation seems to appear at lower conversions. The fact that the appearance of the first insoluble fraction is temperature independent and percolation is temperature dependent, both phenomena appearing at different stages of the polymerization reaction demonstrates the complex gelation behaviour of this system and suggests that percolation and gelation correspond to two different phenomena linked to the heterogeneous nature of this system. Percolation should correspond to the macroscopic behaviour of the system, while gelation must be restricted to the microscopic domain. 

One of the most common methods for studying compositional heterogeneity in multicomponent systems is the measurement of glass transition temperature [43]: the presence of more than one T_g_ can generally be interpreted as an indication of heterogeneity. The evolution of the glass transition temperature during the curing process of this system was followed at 40 °C and 90 °C and it is shown in Figure 1.

As presented in Table 1, the T_g_ of the pure precursors are −68 °C (PAMS) and −41 °C (HDGEBA), respectively. At the beginning of the reaction, two T_g_ are observed at −55 °C and at −39 °C, demonstrating the biphasic nature of this system. The highest T_g_ is attributed to an HDGEBA-rich phase due to the similarity of the T_g_ values, and remains almost constant throughout the curing process at 40 °C; however, if curing is performed at 90 °C, except at the beginning of the reaction, the system appears to be homogeneous at 90 °C, presenting a single T_g_ that increases with conversion up to 83 °C when the system reaches the limiting conversion of 0.99 (Table 2). 

The lowest T_g_ is attributed to a PAMS rich phase, although it is slightly higher than the T_g_ of pure PAMS. This difference may be due to the partial solubilization of the epoxy component in the polysiloxane component, although pressure effects associated with the different expansion coefficients should not be excluded. Using the Fox equation, it is simple to find that the weight fraction of the epoxy component in this PAMS-rich phase is $w_{H}^{P}=0.5$. At 40 °C, the T_g_ of the PAMS-rich phase increases as curing proceeds up to the limiting conversion of 0.60. For comparison, the amount of conventional DGEBA in the PAMS-rich phase is much lower, $w_{DGEBA}^{P}=0.13$ [27], reflecting the lower solubility of DGEBA in PAMS. 

Concerning the solubilization of PAMS in HDGEBA, the difference of glass transition temperatures between pure HDGEBA and HDGEBA in the curing system is only 2 °C and the application of the Fox equation fails, probably because the amount of dissolved PAMS is negligible. 

A consequence of the partial solubilization of the epoxy in PAMS that may be relevant for simulation studies is that the real volume fraction of PAMS-rich domains will be different than the nominal one; the latter is $\phi_{D}^{N}=0.22$, while the other calculated from the Fox equation is $\phi_{D}^{Fox}=0.44$. Nevertheless, the Fox equation only gives good results for miscible systems, which is not the case here. In fact, the value of 0.44 is much higher than the critical volume fraction or percolation threshold for packing soft spheres, which is about 0.28–0.30 [44] and we have not observed (at a depth of 100 μm) any percolated structure at the beginning of the curing reaction. Experimental PAMS volume fraction must have a value higher than 0.22 but lower than 0.30 and its determination would require the use of complementary techniques or molecular dynamics simulation methods. Current ongoing research is facing this problem.

The above-mentioned experimental facts suggest that: (i) although HDGEBA and PAMS are not miscible, a certain fraction of the epoxy dissolves in the polysiloxane; and (ii) the curing reaction takes place in the siloxane-rich phase, which remains phase separated for curing temperatures near and below 40 °C. Consequently, the morphology of this system at 40 °C would consist of polysiloxane-rich domains dispersed in an almost pure epoxy phase that acts as a matrix. As will be demonstrated later, the domains grow through two mechanisms: coalescence and inwards diffusion of epoxy, the latter at the expense of the epoxy-rich matrix. A scheme of the proposed growth mechanism can be found in the Appendix A. Additionally, these results are coherent with our previous assumption that gelation must be restricted to the PAMS-rich domains and the interconnection of PAMS domains must be associated with the percolation of the system. In-situ observations using laser scanning confocal microscopy will reveal and confirm some morphological details of this process. 

### 3.2. Morphology Analysis of Cured Samples

Images of samples cured up to the limiting conversion at different temperatures from 21 °C to 120 °C were obtained by LSCM (Figure 2). The fluorescence is only exhibited by the labelled hardener, appearing as green domains in the images. Instrumental conditions (gain and offset) were selected in each image to obtain maximum contrast between phases, and the digital resolution of the photomultiplier signal was selected to be 12 bit, so the maximum difference between the brightest and darkest regions is about 4 × 10^3^ digital units. Therefore, only relative intensity comparisons can be made between the images. 

The morphology of the system at low curing temperatures is similar to the previously reported DGEBA/PAMS reactive system [27,45] and consists of a dispersion of quasispheroidal siloxane-rich domains in an epoxy-rich matrix. At low curing temperatures, domain sizes are in the order of 50 μm. At higher temperatures, domains become fuzzier, suggesting the presence of wide and diffuse interphases between PAMS-rich and epoxy-rich regions. At 90 °C and above, the cured system seems to be microscopically homogeneous. The main differences between the morphology of DGEBA/PAMS [45] and HDGEBA/PAMS cured samples are: (i) domain size, which is appreciably larger for the system based on the hydrogenated resin, probably due to its lower viscosity ($\eta_{DGEBA}=3.2\;Pa\cdot s$); and (ii) homogeneity of samples cured at 90 °C and above, reflecting a better solubility of HDGEBA than DGEBA in PAMS, since the latter system remains heterogeneous even if cured at 120 °C. To illustrate these differences, Figure 3 shows the limiting interphase thickness and compositional gradient for the samples presented in Figure 2. 

The interphase thickness and compositional gradient were determined from intensity–distance profiles across PAMS-rich domains, as explained in Appendix A. Data for the DGEBA/PAMS system [45] are provided in this Figure for the purpose of comparison. At low curing temperatures, interphase thickness is appreciably higher and the compositional gradient is appreciably smaller for HDGEBA than observed for DGEBA/PAMS. As curing temperature increases, the interphase gradient decreases for both systems, but from 80 °C on, the HDGEBA system becomes almost homogeneous, while DGEBA remains heterogeneous. Concerning the interphase width, it becomes fixed at about 15 μm for DGEBA, although it continues to increase for HDGEBA. These results are determined by the previously suggested mechanism for our system: gradient decreases and interphase thickness increases because the outer epoxy fluid diffuses through PAMS domains, where it reacts and becomes chemically bound, and coalescence of PAMS domains may drag some of the epoxy that surrounds the surface of the parent domains inside the daughter domain. Evidence for coalescence is given in the next section.

### 3.3. Morphology Evolution with Conversion at 40 °C

To understand the processes which generate the morphology in reactive hybrid HDGEBA/polysiloxane blends, the curing of the labelled HDGEBA/PAMS system was monitored “in situ” by LSCM at 40 °C. Images taken at different times (or conversions) are shown in Figure 4. This low curing temperature was chosen because the reaction is slow enough to enable the capture of high-resolution images at low conversions.

Figure 4 shows that below microgelation, quasispheroidal PAMS-rich domains grow as conversion increases due to the inward diffusion of the epoxy matrix and due to coalescence. The figure also shows two coalescence events: (a,b) and (c,d). Event (a,b) was captured before microgelation where local microviscosity should be low enough to enable the almost spherical shape to be recovered. This observation is coherent with the low glass transition temperature of the PAMS-rich domains before microgelation (Figure 1), which is 60 °C below the curing temperature. Interestingly, above microgelation, when crosslinking density becomes high, recovering the spheroidal shape is no longer possible, as reflected by event (c,d): composition redistribution yields tubular-like domains that may grow throughout the reacting mass. The growth of this kind of object may be the microscopic origin for percolation as measured by viscometry.

For quantitative image analysis, the number of domains and their average diameter were measured and plotted against conversion in Figure 5. In the early stages, there are many small domains, but as the reaction proceeds, the number of domains decreases in a sigmoidal fashion. Above microgelation $\left(\alpha \geq 0.15\right)$, the number of domains is only 7% of its initial value, remaining constant up to the limiting conversion. The average diameter increases with a conversion of about 350% up to microgelation and only increases an additional 14% above it. Variations of interphase thickness and compositional gradients (Appendix A) also confirm that microgelation slows down or even stops the evolution of morphology. This observation is in accordance with what should be expected from a gelled system: at 0.15 conversion, PAMS-rich domains gel is no longer able to coalesce in the form of spherical particles. It is interesting to highlight that, contrary to what should be expected in a purely stochastic process, there appears to be an induction period in the decrease rate of the number of domains, i.e., the maximum rate appears at about 5% conversion. The same induction period appears for the rate at which size increases.

For systems where particle surface is a control parameter that governs mass transfer, as in our case, it is acceptable to describe the size distributions by the Sauter diameter [46] or third moment of the size distribution, defined as ${<d>}_{32}=\sum_{i}n_{i}d_{i}^{3}/\sum_{i}n_{i}d_{i}^{2}$. The plot of this average as a function of epoxy conversion is presented in Figure 5 (bottom). This average weighs large particles more heavily than small ones and reveals coalescence events in a much clearer fashion than the simple average. The dotted lines suggest that the probability of coalescing events is, obviously, not uniform along the curing reaction because of the relatively small number of domains. It appears from the eye guides in Figure 5 that, starting with an average diameter of about 20 μm, the probability of forming dimers reaches maximum at about 5% conversion, trimers at about 9% conversion, and higher order aggregates at higher conversions.

### 3.4. 0-Time Distribution

A detailed inspection of the diameter distribution at the beginning of the reaction (see Appendix A) shows that it is not centred at a single value around the average. It appears to be a multimodal distribution, which may be a clear consequence of the emulsification system used in this work. The process of emulsification can be described as follows: when mixing the two fluids with the magnetic stirrer (see Experimental part), the PAMS phase becomes dispersed in the form of droplets inside the HDGEBA matrix. Under the mixing geometry and conditions used, the maximum Reynolds number is in the range of 0.2–0.6 [47], indicating that mixing occurs in a laminar flow. In laminar flows, the viscous epoxy matrix exerts shearing stresses to the spherical PAMS-rich droplets deforming them; when opposing these forces, surface forces arising from the interfacial surface tension keep the integrity of the droplet. For viscosity ratios (dispersed/matrix) in the range 0.1–1 (our case is 0.4), the shearing force may reach enough high values to induce droplet burst [46] either in the form of small droplets through the so-called tip streaming phenomenon, or into two daughter-droplets with small satellite droplets [48]. A concomitant process is droplet coalescence, which may be favoured in regions of the stirring system where droplet break-up is not predominant [47], i.e., in regions where the shear rate is small. Therefore, the resulting emulsion formed in the mixing stage, at α~0, will consist of newly formed droplets with varying size, along with dimers, trimers, etc., and this may be the explanation of the multimodal size distribution presented in Appendix A.

This distribution becomes fixed under the subsequent quiescent conditions inside the microscope stage and starts to grow in the absence of external forces. With the current experimental setup and mixing system, it is technically unfeasible to recover the original monomodal size distribution. Therefore, for the present discussion, we consider the first image obtained as our 0-time distribution (see Figure 4, α=0.01, which is almost the same as the image for α=0). 

### 3.5. Simulation of the Coalescence and Growth of PAMS-Rich Domains

Since the early theory of M. Smoluchowski [49], the majority of models consider coalescence as a stochastic process by which two (mother) particles diffuse through a solvent medium, encounter each other, and merge into a single (daughter) particle [50]. Using the well-known Stokes–Einstein equation for a spherical particle, it is possible to estimate the diffusivity of a PAMS-rich domain as $D=\frac{k_{B}T}{6\pi \mu r}\approx 10^{-17}\frac{m^{2}}{s}$; as a characteristic domain diffusion time, we can consider the time needed to travel a distance equivalent to its radius, which is $t=\frac{\langle x^{2}\rangle }{2D}\approx 10^{9}\;s$. As can be observed in Appendix A, the time span in which morphology becomes fixed (up to α=0.15) is six orders of magnitude lower, about 2000 s. Consequently, coalescence in this reactive system is not driven by diffusion of the PAMS-rich domains. The 0-time distribution remains essentially fixed in space all along the process. Therefore, domains can only grow by inwards diffusion of the epoxy matrix; as domains grow, they get closer and closer until the epoxy film around them breaks and coalesces [47]. To our knowledge, this specific coalescing mechanism has not been considered in current theories, but it is suitable for simulation with the following assumptions:(i)The diffusion rate of epoxy fluid through the domains must be almost the same as the reaction rate. This assumption seems to be rather stringent, but it is based on experimental observations. If the diffusion rate is much higher than the reaction rate, after some induction period, the domains would grow independently of conversion and this has not been observed (see Figure 5). On the other hand, if the diffusion rate is lower than the reaction rate, the reaction becomes diffusion-controlled and the rates of diffusion and reaction of the HDGEBA molecules will be equal.(ii)The spherical shape of the particles remains after collisions. This is a common assumption in all coalescence models [49,50] and it has also been observed experimentally (see Figure 4) before microgelation. Consequently, once an encounter takes place, the volume of the daughter particle is the sum of the volumes of the parent molecules, and its centre of mass changes accordingly. An encounter takes place when the distance between two particles is equal to the sum of their radii.(iii)Initial PAMS-rich domains consist only of pure PAMS. This assumption appears to be contradictory to the observed glass transition temperature of the PAMS phase in the initial mixture. Since the volume fraction of PAMS-rich domains is uncertain, we will explore some other values of the initial volume fraction keeping constant the equivalent ratio.(iv)The initial distribution of sizes is taken as a monomodal Gaussian distribution centred around a given average radius with a standard deviation and upper and lower cut-offs as observed experimentally. This assumption excludes both very small and very big particles that have not been observed experimentally (see Appendix A). Several initial average sizes will be explored to analyse their influence on the coalescence process.

A detailed explanation of the algorithm used for the preparation of the initial 3D distribution of domains in a box of volume $V_{T}$, their growth with conversion, and their coalescence is presented in the Appendix A. However, the most relevant equations are summarized here. 

As chemical conversion α increases, domains start to grow. When conversion changes from $\alpha_{k}$ to $\alpha_{k+1}$, volume fraction changes as $\Delta \phi \left(\alpha \right)=\phi \left(\alpha_{k+1}\right)-\phi \left(\alpha_{k}\right)=\left[1-\phi_{D}^{N}\right]\cdot \left[\alpha_{k+1}-\alpha_{k}\right]$, where $\phi_{D}^{N}$ is the initial volume fraction of domains. Calling $\delta \alpha =\left[\alpha_{k+1}-\alpha_{k}\right]$, the volume of epoxy component that enters the domains is $\Delta V=V_{T}\left[1-\phi_{D}^{N}\right]\delta \alpha$. This volume is distributed among all the particles of the box according to their surface since mass transfer occurs through the surface of domains. Consequently, the volume of each particle, $V_{i}$, will increase an amount $\delta V_{i}$ given by:$$\delta V_{i}=V_{T}\left[1-\phi_{D}^{N}\right]\delta \alpha \cdot \frac{S{\left(\alpha_{k}\right)}_{i}}{\sum_{i=1}^{n}S{\left(\alpha_{k}\right)}_{i}}$$

The volume of the particle when the system has experienced a conversion change δα will now be $V{\left(\alpha_{k+1}\right)}_{i}=V{\left(\alpha_{k}\right)}_{i}+\delta V_{i}$, and from this new volume the corresponding size of each domain can be easily calculated.

Considering periodic boundary conditions, the distance between a particle and its neighbour $d{\left(\alpha_{k+1}\right)}_{ij}$ is calculated. If this distance meets the condition $d{\left(\alpha_{k+1}\right)}_{ij}\leq R{\left(\alpha_{k+1}\right)}_{i}+R{\left(\alpha_{k+1}\right)}_{j}$, where $R_{i},R_{j}$ are the radius of the parent domains, a coalescent event will occur forming the daughter particle *m*. The volume of the new particle will be the sum of the volumes of the parent *i,j* particles; the new radius, $R_{m}$, can be calculated assuming *m* is a sphere; and the coordinates of the centre of the new domain will be the coordinates of the centre of mass. Table 3 shows the main parameters that have been varied in the simulations (an extended Table can be found in the Appendix A).

Figure 6 and Figure 7 present some selected findings of the simulations along with a comparison with experimental results.

In Figure 6, the normalized number of domains as a function of conversion is compared with experimental results. The slopes of the curves are a measure of the coalescence rate. On the left, the initial radius is fixed at 10 μm and the domain volume fraction is varied between 0.16 and 0.3; on the right, the volume fraction is fixed at 0.2 and the initial radius is varied between 8 and 20 μm. Normalization was necessary to compare with experimental results because the initial number of particles in the simulations was much higher than in experiments. The conversion was restricted to about 20% because images show that above 15%, morphology evolution was stopped. Comparison of both figures shows that the initial size of the domains has almost negligible influence on the coalescence rate, being the initial volume fraction the most important variable. It can be observed that the rate of coalescence is at its maximum at the beginning of the process for the volume fraction values explored in this work ($0.16<\phi_{D}^{N}<0.3$), and then progressively slows down until conversion reaches 15%, where the number of domains appears to level-off. We attribute this behaviour to the proximity of the explored volume fraction values to the percolation threshold; the initial box contains a huge number of particles in close contact, and as conversion starts to increase, particles coalesce immediately. 

The good coincidence of the simulations with experimental results in the range 0.05≤α<0.15 for initial volume fractions around 0.2–0.25, near the nominal initial volume fraction is notable; this finding is in accordance with what was expected from the analysis of the glass transition temperatures. However, below 5% conversion, experimental results show a clear induction period that is absent in the simulations, i.e., the experimental coalescence rate is smaller than predicted by the simulations at the beginning of the reaction. 

This behaviour could be explained if part of the HDGEBA had entered the PAMS phase during the emulsification stage, i.e., there is a partial solubility of the resin into the polyaminosiloxane. Some experimental facts support this hypothesis: the initial T_g_ of PAMS-rich phase is higher than that of pure PAMS, and qualitative analysis of the LSCM images at the beginning of the reaction (see Figure 4) suggest a volume fraction of the PAMS-rich phase higher than the theoretical (surface analysis of the PAMS-rich phase is between 0.23 and 0.30, depending on the cut-off selected). If part of the HDGEBA is present in the PAMS-rich phase from the beginning, the reaction would occur both in the interphase and the bulk. It is reasonable to assume a faster reaction rate for HDGEBA molecules inside the domains with respect to the ones at the interface because they are completely surrounded by PAMS molecules. That could bring the observed induction period, but, interestingly, the simulation also shows that the amount of HDGEBA partially solubilized in PAMS must be very low. A simple calculation shows that the amount of HDGEBA necessary to have an induction period up to α=0.02 would suppose only a 7% *w*/*w* of HDGBA in the initial PAMS-rich phase, increasing its volume fraction from 0.22 to only 0.24, compatible with the experimental results. 

It Is not possible to introduce this variable in the simulations without knowing the reactivity ratio between HDGEBA at the bulk and at the interphase, but, assuming a higher reactivity at the bulk, the reservoir of resin inside the PAMS-rich phase will be consumed in the early reaction stages, and subsequent reaction progress will be due only to reaction/diffusion at the interphase. This induction period, probably caused by partial solubility, does not enter into conflict with the simulation approach.

Simulations of the variation of the normalized average size $\langle R\rangle /{\langle R\rangle }_{0}$ and the normalized Sauter diameter with conversion for several initial domain volume fractions $\phi_{D}^{N}$ at fixed domain size (10 μm) are compared with the experimental results in Figure 7. It can be observed that the simulations can capture the most relevant features of the growth process: (i) the increase of both domain size and third moment of the size distribution; and (ii) the plateau in the variation of 〈R〉 with conversion. However, this figure also shows that experimental data departs from predictions, most likely because simulation assumptions simplify the real behaviour of this complex system.

For example, a detailed analysis of Figure 7 shows that experimental $\langle R\rangle /{\langle R\rangle }_{0}$ values increase more steeply than simulations (for α>0.05). The initial amount of HDGEBA in PAMS-rich domains is governed by the thermodynamic interaction between the highly self-associated polyaminosiloxane and the slightly polar epoxy resin. It seems plausible that once some HDGEBA fragments have been inserted in the PAMS chains, interactions between both components will be progressively more favourable and more HDGEBA will enter the domains without the need for a chemical reaction. This means that the PAMS-rich domains may grow with a small increase of conversion as the experimental data reflects. 

Concerning point ii, it is noteworthy to observe that simulations predict a plateau in the size variation with conversion, as shown by the experimental data (see also Figure 5). This plateau appears at a conversion that depends on the initial volume fraction $\phi_{D}^{N}$, as depicted by the arrows in Figure 7 (it also depends on ${\langle R\rangle }_{0}$, but this is not shown). Microgelation appears at α=0.15 and simulations show that, for initial volume fractions in the range 0.2–0.25, near the nominal ones, a plateau also appears at α=0.15. However, in our simulations, we have not incorporated any mechanism that allows the calculation of gelation. In fact, our simulation is purely geometric and, since particles must grow at any time by diffusion/coalescence, we must conclude that microgelation and the presence of a plateau is nothing more than a coincidence.

## 4. Conclusions

The hybrid HDGEBA/PAMS reactive system has been studied under stoichiometric conditions by FTnIR, DSC, gel point, viscometry, and LSCM to understand how the morphology becomes fixed. FTnIR allowed us to obtain time–conversion master curves at several temperatures. DSC results show that the system was biphasic: pure HDGEBA and PAMS-rich phases. The exact volume fraction of the latter was uncertain: it was higher than the nominal, stoichiometric one (0.22), and lower than the critical volume fraction for percolation (~0.3). Gel point conversion as determined by gravimetry was 0.15, somewhat lower than the prediction for homogeneous systems (0.23); however, viscometry showed that percolation appeared at ~0.30 conversion, very near to the predictions for the percolation of mixed soft spheres. 

Images obtained by in-situ LSCM allowed the determination of the average size of PAMS-rich domains, the number of domains, the third moment of the size distribution, and the variation of these parameters with epoxy conversion. It was found that domains size increases with conversion from 22 μm up to ~80 μm at α=0.15, and the number of domains decrease from ~300 down to ~25 at α=0.15; above this conversion, a plateau is observed in both the size and the number of domains.

These results suggest the following mechanism for morphology evolution: (i) droplets of a PAMS-rich phase were dispersed in a pure HDGEBA matrix after quick mixing both reactive components under laminar conditions; (ii) due to the high viscosity of both phases, the 0-time distribution of PAMS-rich domains became fixed in space and remained fixed all along the curing process; (iii) PAMS-rich domains grow by inwards diffusion of the epoxy matrix; and (iv) as domains grow, they get closer and closer until the epoxy film around them breaks, merges, and forms a new daughter particle. 

This apparently simple mechanism was simulated through Metropolis-Monte Carlo, exploring a wide range of values of the “a priori” relevant variables: initial volume fraction of PAMS-rich domains and initial domain size. These simulations were able to capture the essential features of the morphology evolution process with a reasonable agreement with experimental data. However, it became evident that the complete process is more complex than the geometrical approach of the simulation. Specifically, the main deviations that we have found are: (i) the induction period in the rate of coalescence was absent in the simulations, which could only be explained if both components, epoxy and PAMS, were partially miscible; (ii) the average domain size increases faster than predictions, probably because when HDGEBA starts to be incorporated into the PAMS-rich domains, the miscibility of both components increases. Items (i) and (ii) are connected to the specific interactions between the two components and how these vary with molecular weight, branching, and appearance of secondary and tertiary amines. In this context, molecular dynamic simulations and coarse-grained methods may help to completely elucidate the mechanism of morphology development in these hybrid thermosets.

## Figures and Tables

**Figure 1 polymers-14-05375-f001:**
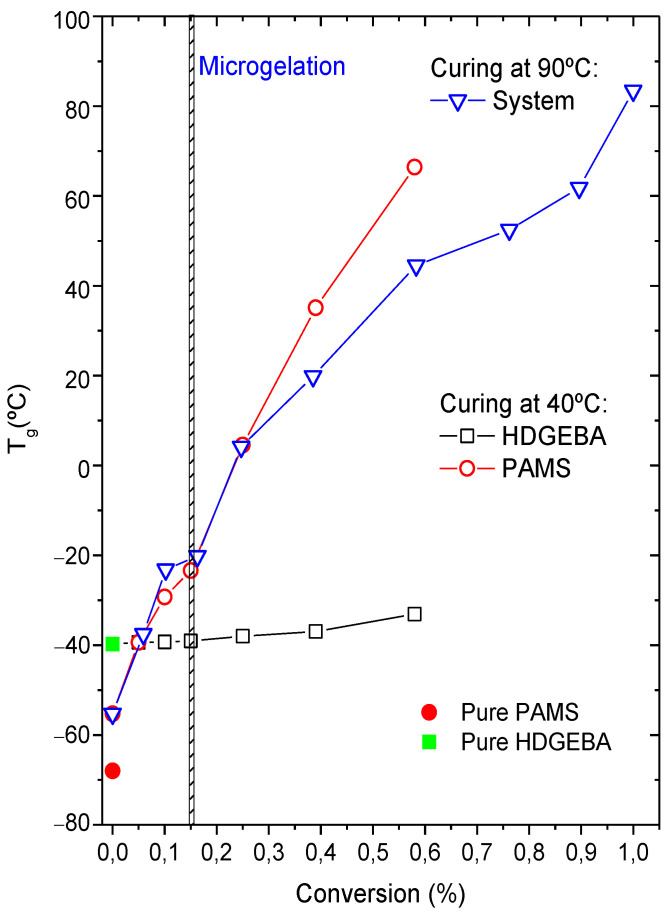
Glass transition temperatures of pure HDGEBA and PAMS and the reactive HDGEBA/PAMS system as a function of epoxy conversion.

**Figure 2 polymers-14-05375-f002:**
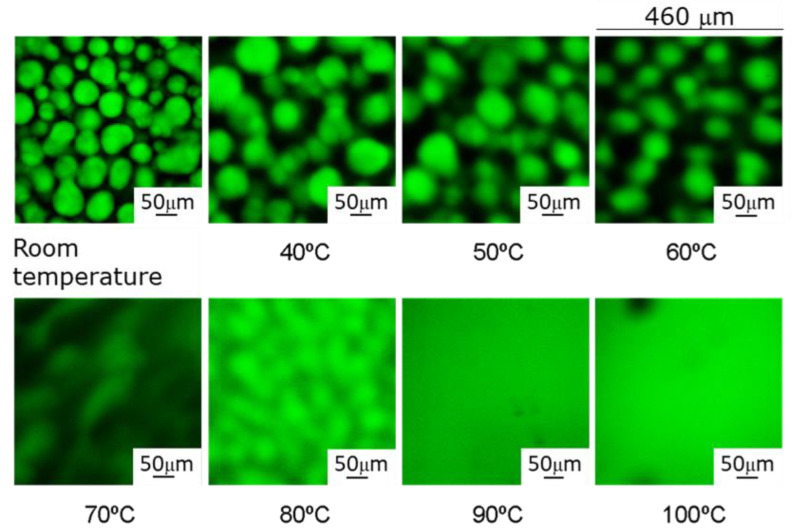
LSCM images of HDGEBA/PAMS cured up to the limiting conversion at different temperatures between room temperature and 100 °C. Objective: 20×. Scale bar 50 μm; scanned area 460 × 460 μm^2^.

**Figure 3 polymers-14-05375-f003:**
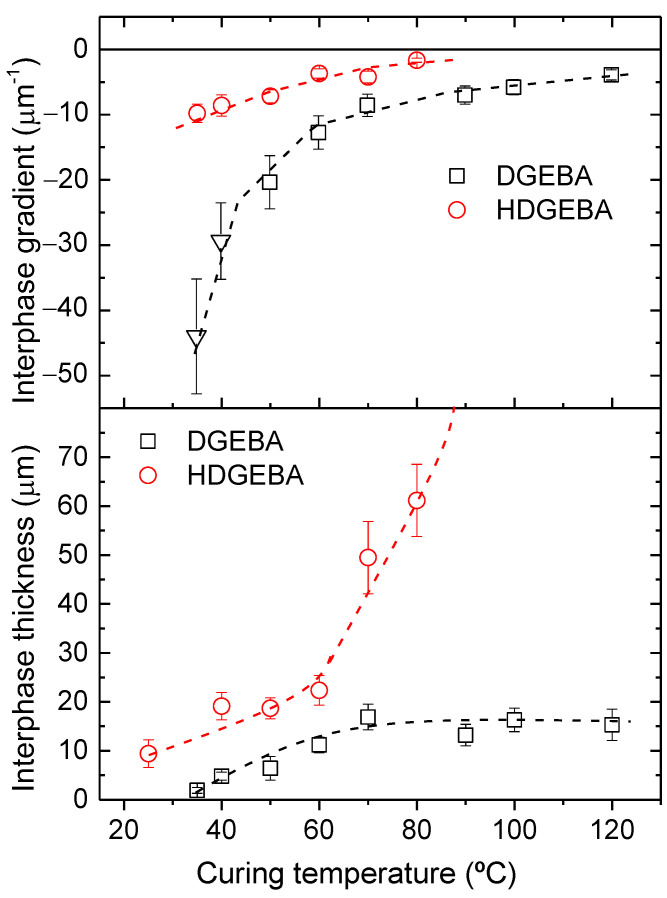
Compositional gradient and interphase thickness for samples containing DGEBA and HDGEBA cured up to the limiting conversion at several temperatures. Error bars correspond to the standard deviations of 10 different images at each curing temperature. Lines are eye-guides.

**Figure 4 polymers-14-05375-f004:**
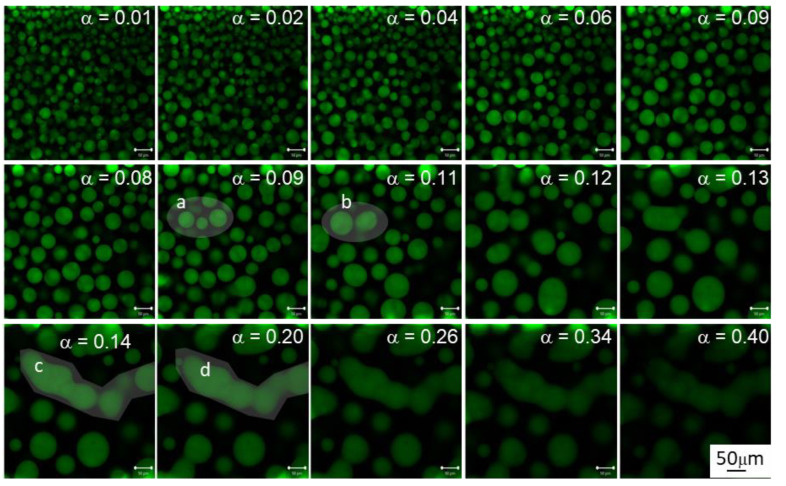
Evolution of HDGEBA/PAMS system morphology at several epoxy conversions (α) at 40 °C. Inserts (**a**,**b**) and (**c**,**d**) correspond to two coalescence events. The scale bar is 50 µm.

**Figure 5 polymers-14-05375-f005:**
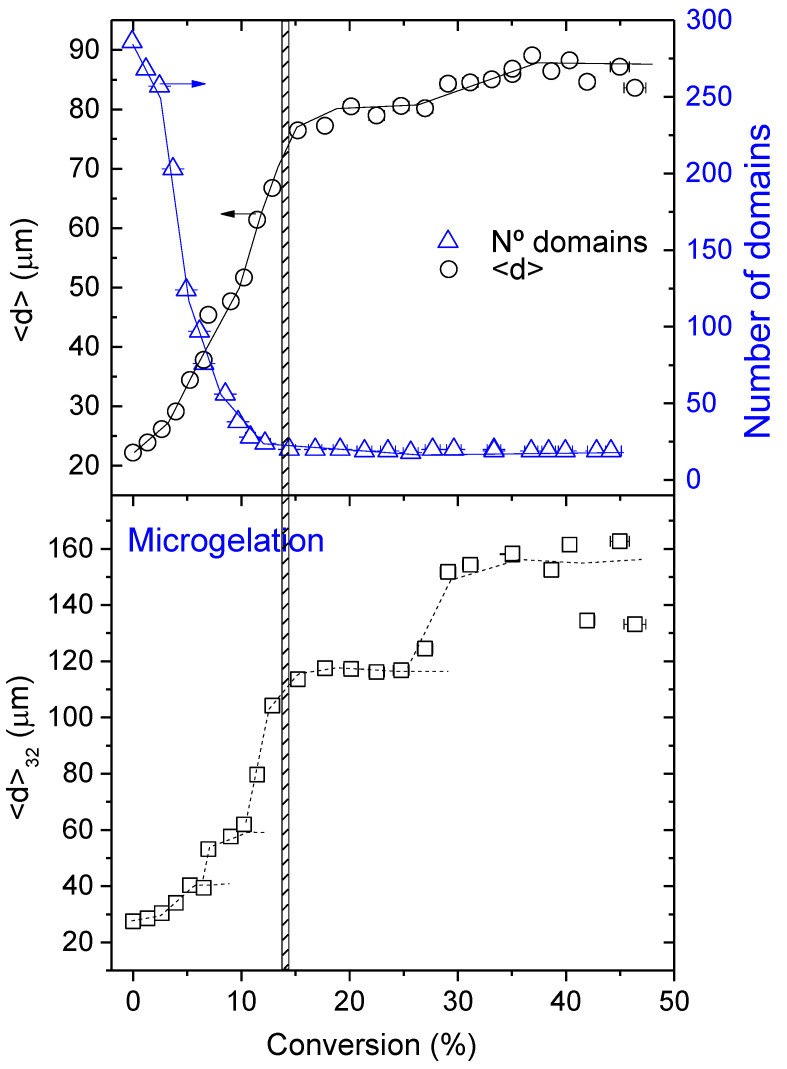
Domain average diameter (<d>), number of PAMS-rich domains, and third moment of the domain diameter distribution $\left({<d>}_{32}\right)$ as a function of epoxy conversion for the system HDGEBA/PAMS at 40 °C. Lines are eye-guides.

**Figure 6 polymers-14-05375-f006:**
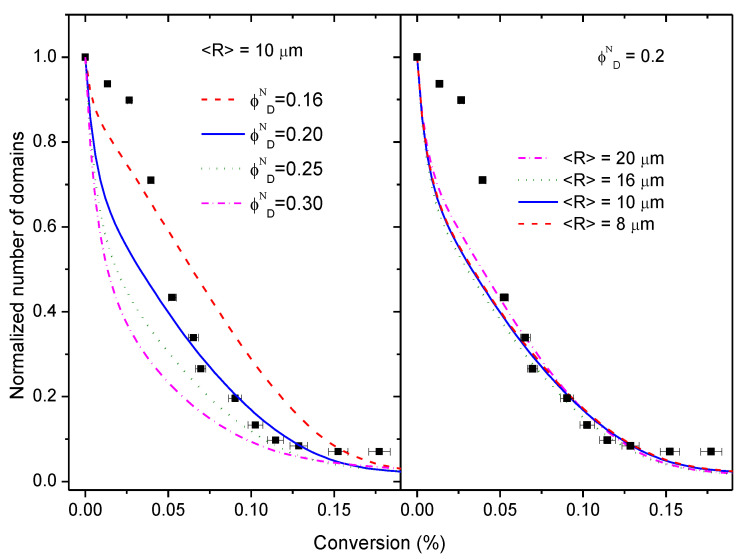
Normalized number of domains as a function of conversion: lines are simulations, square points are experimental data. Left: fixed initial radius of domains at 10 μm and simulations for several initial domain volume fractions. Right: fixed initial domain volume fraction at 0.2 and simulations for several initial domain radii.

**Figure 7 polymers-14-05375-f007:**
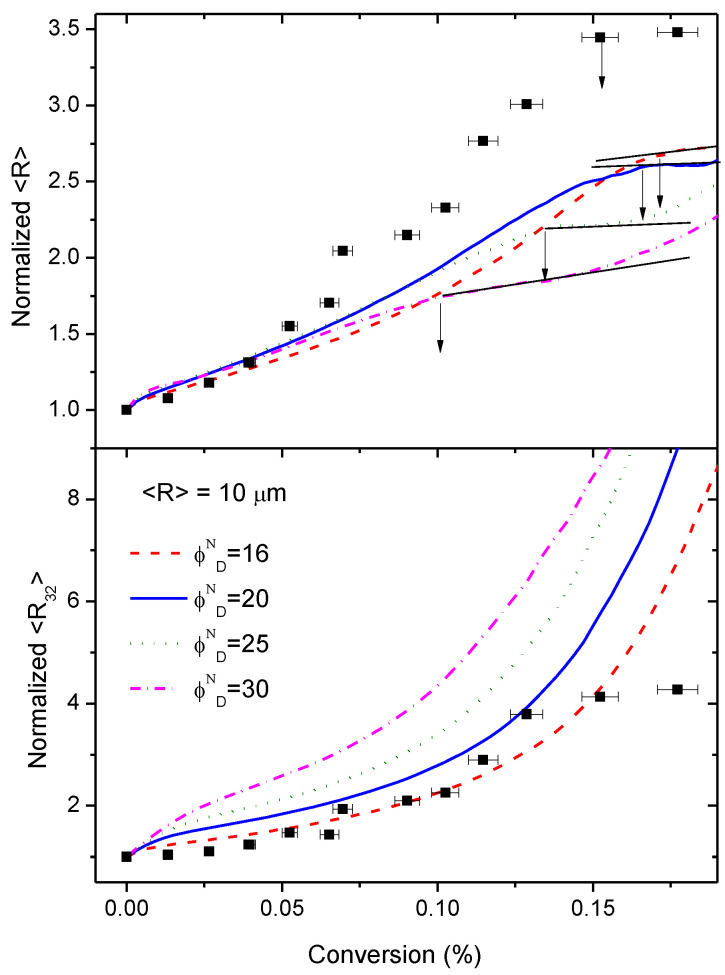
Normalized average diameter, $\langle R\rangle /{\langle R\rangle }_{0}$, (up) and Sauter diameter, $\langle R_{32}\rangle /{\langle R_{32}\rangle }_{0}$ (bottom) as a function of conversion for several initial domain volume fractions fixing the initial average size in 10 μm. Square points are experimental data. Continuous black lines (up) show the plateau in average diameter; arrows show the conversion at which the plateau starts.

**Table 1 polymers-14-05375-t001:** Chemical formulae, weight average molecular mass and functionality, glass transition temperature, density, and viscosity of precursor compounds.

Compound	HDGEBA	PAMS
	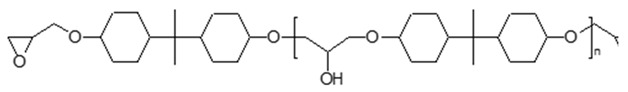	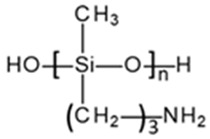
$M_{W}$ $\left(g/mol\right)$	411	1190 ± 150
$T_{g}\;({}^{\circ}C){\;}^{1}$	−41	−68
$f_{W}$	2	20.3 ± 2.7
$\rho \;(g/cm^{3}){\;}^{2}$	1.078	1.055
$\mu \;(Pa\cdot s){\;}^{2}$	1.3	0.53

^1^ see Appendix A; ^2^ at 25 °C.

**Table 2 polymers-14-05375-t002:** Limiting conversion ($\alpha_{lim}$), percolation time $\left(t_{perc}\right)$, and epoxy conversion at percolation $\left(\alpha_{perc}\right)$ at different curing temperatures.

Temperature (°C)	$\alpha_{lim}$	$t_{gel}$ (min)	$\alpha_{gel}$	$\;t_{perc}$ (min)	$\alpha_{perc}$
40	0.60	29.6±1	0.15±0.01	66.7	0.30
60	0.90	12.6±0.5	0.17±0.01	19.4	0.28
70	0.93	8±0.5	0.17±0.01	10.5	0.23
80	0.97	-	-	-	-
90	0.99	-	-	-	-

**Table 3 polymers-14-05375-t003:** Initial volume fraction of PAMS-rich domains $\phi_{D}^{N}$, average domain radius 〈R〉, standard deviation of the initial Gaussian distribution of domains σ, and lower and upper cut-offs of the size distributions.

Run	$\phi_{D}^{N}$	$\langle R\rangle \;\left(\mu m\right)$	$\sigma \;\left(\mu m\right)$	$L_{R}$	$U_{R}$
1–8	14–40	8.0	2.5	2	16
9–16	14–40	10.0	2.5	4	18
17–24	14–40	12.0	2.5	6	20
25–32	14–40	14.0	2.5	8	22
33–40	14–40	16.0	2.5	10	24
41–48	14–40	18.0	2.5	12	26
49–56	14–40	20.0	2.5	14	28

## Data Availability

Data is available from authors on request.

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
