# Peer review of "Mechanism of Morphology Development in HDGEBA/PAMS Hybrid Thermosets: Monte Carlo Simulation and LSCM Study"

_polymers, 2022, doi:10.3390/polym14245375_

Round 1

Reviewer 1 Report

It is an interesting research concerning HDGEBA/PAMS hybrid thermoset, based on your great experience on that systems using confocal microscopy since 2005 (see reference 45). The domains morphology evolution was investigated with: laser scanning confocal microscopy (LSCM) – which is excellent: obtained, presented and discussed; thermal behavior was monitored by DSC spectroscopy; and chemical bonds evolution was monitored by FTIR spectroscopy. The Monte Carlo simulation was considered as results discussion and as novelty element. Unfortunately, DSC and FTIR results were not presented in the manuscript (they might appear in supplementary materials, which are not available). Therefore, the conclusion is not sustained by the experimental results. There are several major aspects to be improved, see to the comments below:

Comment 1) I was not able to see the supplementary materials, the provided link (manuscript line 540) returns the message ,,Error 404 - File not found”. Supplementary materials were not loaded on the submission platform to be available for reviewers. Please make supplementary materials available for reviewers.

Comment 2) Manuscript title is too general. I suggest to improve it in the following manner: ,,Mechanism of morphology development in HDGEBA/PAMS hybrid thermosets: Monte Carlo simulation and LSCM study”.

Comment 3) It is not clearly explained which is the investigated composition. Which was the HDGEBA:PAMS ratio in the investigated sample? How many compositions (different ratio) were investigated? It looks like a single HDGEBA:PAMS ratio was observed in different conditions of temperature and reaction time.

Comment 4) DSC spectra are missing. Perhaps these are presented in the supplementary materials, which are not available. However, it is mandatory to present DSC spectra in the manuscript at the results section to sustain the discussion and conclusions.

Comment 5) FTIR spectra are missing. Perhaps these are presented in the supplementary materials, which are not available. However, it is mandatory to present FTIR spectra in the manuscript at the results section to sustain the discussion and conclusions.

Author Response

Polymers

Lionel Lin, Editor

Leganés, December 2nd, 2022

Dear Editor, enclosed please find our point-by-point response to reviewers’ comments

Reviewer #1

Comment 1) I was not able to see the supplementary materials, the provided link (manuscript line 540) returns the message ,,Error 404 - File not found”. Supplementary materials were not loaded on the submission platform to be available for reviewers. Please make supplementary materials available for reviewers.

Answer 1: Polymers editorial crew made available the supplementary materials document

Comment 2) Manuscript title is too general. I suggest to improve it in the following manner: ,,Mechanism of morphology development in HDGEBA/PAMS hybrid thermosets: Monte Carlo simulation and LSCM study”.

Answer 2: we appreciate the suggestion of reviewer. Title has been changed accordingly

Comment 3) It is not clearly explained which is the investigated composition. Which was the HDGEBA:PAMS ratio in the investigated sample? How many compositions (different ratio) were investigated? It looks like a single HDGEBA:PAMS ratio was observed in different conditions of temperature and reaction time.

Answer 3: In lines 107, 108 and 139 of the manuscript it is mentioned that the epoxy/amine equivalent ratio was kept equal to one (stoichiometric conditions) in all cases; the weight ratio was therefore 210:58.5. In almost all thermosets, maximum glass transition temperature and maximum modulus is obtained at an equivalent ratio of one. This is the reason why departure from this ratio has little interest.

Comment 4) DSC spectra are missing. Perhaps these are presented in the supplementary materials, which are not available. However, it is mandatory to present DSC spectra in the manuscript at the results section to sustain the discussion and conclusions.

Comment 5) FTIR spectra are missing. Perhaps these are presented in the supplementary materials, which are not available. However, it is mandatory to present FTIR spectra in the manuscript at the results section to sustain the discussion and conclusions.

Answer 4&5: FTIR spectra in the near range of the system HDGEBA/PAMS was already published in 2012 where band assignments were done (see Figure 7 of ref [4]). In our opinion, lack of novelty precludes its publication. We have included the sentence “See ref [4] for spectra details” in line 143 of the manuscript. However, for the eyes of the reviewer we reproduce the FT(n)IR spectra of this system at several curing times in the enclosed image

Concerning glass transition measurements, we have used the built-in routine of the Mettler equipment to calculate them from the heat flow thermograms. We only keep thermograms of PAMS and HDGEBA as part of the characterization data set applied in our laboratory to new or synthesized compounds. These have been included in the new Figure SM3 B of supplementary materials and a mention to this figure is done as a footnote of Table 1. For curing experiments, once the Tg is obtained the thermograms are discarded and cannot be recovered.

Reviewer 2 Report

1. The immiscibility concern is not only there for polymer systems, but also for other systems like metals (Cu-Fe alloy). As this is the highlight for this paper, I suggest the authors to have a separate systematic paragraph to discuss the issues caused by immiscibility, and I believe the following literatures could help:

[1] High-strength and high-conductivity in situ Cu–TiB2 nanocomposites. S Pan, T Zheng, G Yao, Y Chi, I De Rosa, X Li. Materials Science and Engineering: A 831, 141952

[2] You, Y., Rong, M.Z. and Zhang, M.Q., 2021. Adaptable reversibly interlocked networks from immiscible polymers enhanced by hierarchy-induced multilevel energy consumption mechanisms. Macromolecules, 54(10), pp.4802-4815.

[3] Shuai, C., He, C., Peng, S., Qi, F., Wang, G., Min, A., Yang, W. and Wang, W., 2021. Mechanical alloying of immiscible metallic systems: process, microstructure, and mechanism. Advanced Engineering Materials, 23(4), p.2001098.

2. At least, the authors should add some materials characterization like FI-IR to prove the composition and phase of the mixture.

3. The direct Mento carlo simulation is ok in this paper. But more integrated methods like Molecular dynamics and LAMMPS, etc. are available for simulation. In the conclusion and discussion, the  authors should briefly touch how the simulation works could be improved.

Author Response

Polymers

Lionel Lin, Editor

Leganés, December 2nd, 2022

Dear Editor, enclosed please find our point-by-point response to reviewers’ comments

Reviewer #2

Comment 1. The immiscibility concern is not only there for polymer systems, but also for other systems like metals (Cu-Fe alloy). As this is the highlight for this paper, I suggest the authors to have a separate systematic paragraph to discuss the issues caused by immiscibility, and I believe the following literatures could help:

[1] High-strength and high-conductivity in situ Cu–TiB2 nanocomposites. S Pan, T Zheng, G Yao, Y Chi, I De Rosa, X Li. Materials Science and Engineering: A 831, 141952

[2] You, Y., Rong, M.Z. and Zhang, M.Q., 2021. Adaptable reversibly interlocked networks from immiscible polymers enhanced by hierarchy-induced multilevel energy consumption mechanisms. Macromolecules, 54(10), pp.4802-4815.

[3] Shuai, C., He, C., Peng, S., Qi, F., Wang, G., Min, A., Yang, W. and Wang, W., 2021. Mechanical alloying of immiscible metallic systems: process, microstructure, and mechanism. Advanced Engineering Materials, 23(4), p.2001098.

Answer 1. Thank you very much, we really appreciate your comments. Polymer blends in which at least two polymers are combined to create a new material with different physical properties are analogous to metal alloys. However, the thermodynamic considerations underlying the formation of phase-separated structures in both types of materials are quite different. The entropy of mixing for polymer blends is usually very low so polymer mixtures are typically immiscible and phase separate. In fact, few reports of pairs of polymers present thermodynamic miscibility and most polymer blends are not only immiscible but also mechanically incompatible. This contrasts with metal alloys for which the entropy of mixing is high giving rise to rich and complex solid-state phase mixtures: pure compounds, solid solutions and intermetallics along with diffusion or diffusion-less driven micro and nanostructures. For polymers, the development of a miscible blend depends on many factors that include molecular weight, composition, viscosity ratio, interfacial tension, and intermolecular interactions, that is, a series of parameters different than those that affect metal alloys and their performance.

Following the suggestion of the referee we have added a new paragraph discussing the effects of immiscibility in reactive systems in line 45.

Comment 2. At least, the authors should add some materials characterization like FI-IR to prove the composition and phase of the mixture.

Answer 2. FTIR spectra in the near range of the system HDGEBA/PAMS was already published in 2012 where band assignments were done (see Figure 7 of ref [4]). In our opinion, lack of novelty precludes its publication. We have included the sentence “See ref [4] for spectra details” in line 143 of the manuscript. However, for the eyes of the reviewer we reproduce the FT(n)IR spectra of this system at several curing times in the enclosed image

Comment 3. The direct Mento carlo simulation is ok in this paper. But more integrated methods like Molecular dynamics and LAMMPS, etc. are available for simulation. In the conclusion and discussion, the  authors should briefly touch how the simulation works could be improved.

Answer 3. We appreciate de suggestion of the reviewer. We have included a brief sentence in line 211 and a short paragraph in line 542

Round 2

Reviewer 1 Report

The requested correction and completions were well effectuated. I agree yours explanations presented in the cover letter.